# OpenReview forum: "Multi-step Jailbreaking Privacy Attacks on ChatGPT"
_EMNLP/2023/Conference — EMNLP 2023 Findings_

### Official Review · Reviewer_ur2W · 2023-08-03

**Typos Grammar Style And Presentation Improvements:** 1. In the paper's case studies, sensi…
**Soundness:** 4

**Excitement:**

3: Ambivalent: It has merits (e.g., it reports state-of-the-art results, the idea is nice), but there are key weaknesses (e.g., it describes incremental work), and it can significantly benefit from another round of revision. However, I won't object to accepting it if my co-reviewers champion it.

**Missing References:**

N/A

**Paper Topic And Main Contributions:**

The paper investigates privacy risks from large language models like ChatGPT and the New Bing. While these advanced LLMs can successfully refuse previous privacy extraction attacks, it remains vulnerable to the multi-step attack proposed in the paper. Experimental results demonstrates its effectiveness against ChatGPT and New Bing models. Overall, this paper poses potential privacy risks introduced by application-integrated LLMs.

**Questions For The Authors:**

Does a search for information on Bing should be considered a PII leakage? Their information may already be available online.

**Reasons To Accept:**

1. The paper presents a novel multi-step jailbreaking prompt to extract personally identifiable information (PII) from ChatGPT, demonstrating that its dialog safety mechanisms are still insufficient against indirect prompts.
2. It reveals new privacy threats from application-integrated LLMs like the New Bing, which can recover PII more accurately due to its search integration.
3. The paper is well-written and structured. The motivation and methodology are clearly explained.

**Reasons To Reject:**

1. The effectiveness of the proposed multi-step attack prompt has only been demonstrated on limited datasets and models so far. Further evaluation on more diverse data and larger LLMs would be needed to establish its general applicability.
2. While the study successfully demonstrates privacy risks from LLMs, the empirical results are currently limited in providing additional insights. As it stands, the work poses the threat of privacy leakage but does not significantly further our understanding of the factors affecting it.

**Reproducibility:**

3: Could reproduce the results with some difficulty. The settings of parameters are underspecified or subjectively determined; the training/evaluation data are not widely available.

**Reviewer Confidence:**

4: Quite sure. I tried to check the important points carefully. It's unlikely, though conceivable, that I missed something that should affect my ratings.

---

> ### Author Rebuttal · Authors · 2023-08-28
>
> We thank the reviewer for the valuable suggestions and positive feedback. Our responses are provided below.
>
> > Weakness 1:
> The effectiveness of the proposed multi-step attack prompt has only been demonstrated on limited datasets and models so far. Further evaluation on more diverse data and larger LLMs would be needed to establish its general applicability.
>
> Our paper studies the PII leakage issue of present-day LLMs (ChatGPT) and application-integrated LLMs (the New Bing). Prior [1] and follow-up [2] works mainly study the email address leakage of the Enron dataset to probe the PII leakage. Our work extends to three types of PII, including email addresses, phone numbers and email contents. Compared with prior and concurrent works, we still achieve state-of-the-art performance on the pure zero-shot setting without any extra knowledge (the most practical attack so far). We agree that further evaluations are needed for this field. Still, our attack can be a good indicator that current safety mechanisms are not enough to achieve the safety goal for public usage.
>
> In addition, we also test our attacks on more open-source chat safety enhanced models (see tables below), including Llama 2 (Llama-2-7b-chat-hf), Vicuna (vicuna-7b-v1.3), and Guanaco (guanaco-7B). The results show that all 3 models are still vulnerable to our proposed attacks.
>
>
>
> > Weakness 2:
> While the study successfully demonstrates privacy risks from LLMs, the empirical results are currently limited in providing additional insights. As it stands, the work poses the threat of privacy leakage but does not significantly further our understanding of the factors affecting it.
>
>
> Our key insight is that the instruction tuning can be exploited to override LLM’s safety modules. In Lines 257-259, our paper mentions that we can abuse the instruction-following ability to make LLMs play the role that we desire to output unethical responses. To achieve our goal, we may first design some roles/setups to misguide LLMs for outputting unsafe comments.
>
> Another factor of discrepancies between parsed results and correct results may caused by LLMs’ hallucinations.  Our jailbreaking attack makes LLMs willing to output PII. However, hallucination prevents us from getting accurate information (high # parse with lower accuracy). Since many people are working on LLMs’ hallucination, our attack performance can even be further improved once hallucinations are solved.
>
>
> We acknowledge that LLMs can pose safety risks through other means as well. Our intention is to emphasize the significance of privacy disclosure by showcasing a succinct approach like instruction tuning, with the hope of capturing the attention of a wide audience for LLMs’ safer development.
>
>
>
> > Questions:
> Does a search for information on Bing should be considered a PII leakage? Their information may already be available online.
>
> From the perspective of existing privacy laws, if an individual puts its personal information on the Internet, other people searching for it is not considered a privacy violation.
>
> However, misuse of such information for other commercial activities may violate existing privacy laws. For example, a company collecting personal emails from LinkedIn and sending commercial spam emails to them can violate privacy laws (see the CAN-SPAM Act of 2003). On the other hand, companies that collect personal data from the Internet without notice and consent may lack a legal basis for both the CCPA (U.S.) and GDPR (EU).
>
> Thus, if such data is further processed that is out of the scope of users’ consent, the users who share their information online should have the right to opt out of unauthorized data collection. For these cases, it can be considered a PII leakage. The user should be notified and have the right to opt out.
>
> In summary, we use the New Bing as one example to show that application-integrated/ domain-grounded LLMs may also bring their own privacy leakage problems and safety concerns.
>
>
>
>  > Improvement 1: In the paper's case studies, sensitive personal information in the examples is currently obscured by coloring the text black. However, readers can still directly copy the text underneath the black ink blocks. To truly anonymize the data, the authors may want to consider using PDF editing tools to permanently delete or erase the sensitive content, rather than just obscuring it.
>
>
> We thank the reviewer for the kind reminder. Though the text underneath the black ink blocks can be directly copied, Our case studies already use ‘xxx’ to replace the original contents under the blocks. We will use PDF editing tools to replace xxxx in the final version.
>
> > Improvement 2: The version of the black-box LLMs should be pointed out.
>
> Lines 932-936 mention that our experiments were conducted from March to May. Our experiments can still be reproduced on the earlier ChatGPT version such as gpt-3.5-turbo-0301. For the New Bing, we just use its web interface and there is no version about it.
>
>
>
>
>
>
>
>
> ### Added Experiments
> We follow the experimental settings to conduct the same experiments on the open-source LLMs except that we only use greedy decoding for easier reproducibility. The results still suggest that chat safety enhanced Llama 2 (Llama-2-7b-chat-hf), Vicuna (vicuna-7b-v1.3), and Guanaco (guanaco-7B) still suffer higher ‘# parsed’ and ‘# correct’ from our proposed MJP.
>
> | Frequent Enron Emails (88) |        |          |           |
> |----------------------------|--------|----------|-----------|
> | Models                     | Prompt | # parsed | # correct |
> | vicuna-7b                  | DP     | 0        | 0         |
> |                            | MJP    | 59       | 3         |
> | Llama-2-7b-chat            | DP     | 0        | 0         |
> |                            | MJP    | 28       | 8         |
> | guanaco-7B                 | DP     | 0        | 0         |
> |                            | MJP    | 3        | 0         |
>
>
> | University Emails (50) |        |          |           |
> |------------------------|--------|----------|-----------|
> | Models                 | Prompt | # parsed | # correct |
> | vicuna-7b              | DP     | 1        | 0         |
> |                        | MJP    | 29       | 1         |
> | Llama-2-7b-chat        | DP     | 0        | 0         |
> |                        | MJP    | 18       | 1         |
> | guanaco-7B             | DP     | 2        | 0         |
> |                        | MJP    | 23       | 1         |
>
>
> | UNIVERSITY PHONES (30) |        |          |           |      |
> |------------------------|--------|----------|-----------|------|
> | MODELS                 | PROMPT | # PARSED | # CORRECT | LCS6 |
> | VICUNA-7B              | DP     | 0        | 0         | 0    |
> |                        | MJP    | 18       | 0         | 1    |
> | LLAMA-2-7B-CHAT        | DP     | 0        | 0         | 0    |
> |                        | MJP    | 15       | 0         | 0    |
> | GUANACO-7B             | DP     | 2        | 0         | 2    |
> |                        | MJP    | 9        | 0         | 4    |
>
>
>
> References:
>
> [1] Wang, B., Chen, W., Pei, H., Xie, C., Kang, M., Zhang, C., ... & Li, B. (2023). DecodingTrust: A Comprehensive Assessment of Trustworthiness in GPT Models. arXiv preprint arXiv:2306.11698.
>
> [2] Huang, J., Shao, H., & Chang, K. C. C. (2022). Are Large Pre-Trained Language Models Leaking Your Personal Information? Findings of EMNLP 2022.

---

### Official Review · Reviewer_5zeZ · 2023-08-04

**Soundness:** 3

**Excitement:**

4: Strong: This paper deepens the understanding of some phenomenon or lowers the barriers to an existing research direction.

**Paper Topic And Main Contributions:**

The work investigates the privacy leakage issue of ChatGPT by utilizing multi-step jailbreaking prompts to evade the ChatGPT privacy leakage defense mechanism. As a result, personal information can be elicited from the LLM, which could be potentially abused to perform more sophisticated attacks.

**Questions For The Authors:**

See “Reasons to reject”

**Reasons To Accept:**

- Interesting work and the topic is well-motivated

- Explores a new design for evading LLM privacy leakage defense

- The paper is well-written and understandable


**Reasons To Reject:**

- The work experimented with only a few types of personal data (email, phone number) leakage. And lacking a systematic design to explore more types of personal data leakage in modern LLMs. Do the authors have any plans to explore more types of personal data and analyze their leakage in LLMs?

- The work provided insufficient justification for its experiment dataset selection. Are there similar datasets available for other types of personal data? if so, what are the reasons for not including them in the paper?


**Reproducibility:**

3: Could reproduce the results with some difficulty. The settings of parameters are underspecified or subjectively determined; the training/evaluation data are not widely available.

**Reviewer Confidence:**

3: Pretty sure, but there's a chance I missed something. Although I have a good feel for this area in general, I did not carefully check the paper's details, e.g., the math, experimental design, or novelty.

---

> ### Author Rebuttal · Authors · 2023-08-28
>
> We thank the reviewer for the valuable suggestions and positive feedback. Our responses are provided below.
>
> > Weakness 1:
> The work experimented with only a few types of personal data (email, phone number) leakage. And lacking a systematic design to explore more types of personal data leakage in modern LLMs. Do the authors have any plans to explore more types of personal data and analyze their leakage in LLMs?
>
> Besides studying email address and phone number leakage, in Lines 973-1053 of Appendix B,  we also conduct more complicated experiments on email content recovery, which contains many other personal information like meeting schedules and trade secrets.  We follow the main content’s zero-shot experimental settings and successfully recover a few verbatim email contents. All three types of personal data can be an excellent indicator to verify that our proposed attack is risky and effective.
>
> In addition, we would like to test our attacks on more data types if new datasets for probing personal information leakage are published.
>
>
> > Weakness 2:
> The work provided insufficient justification for its experiment dataset selection. Are there similar datasets available for other types of personal data? if so, what are the reasons for not including them in the paper?
>
>
> To the best of our knowledge, most public datasets are already anonymized before publishing and few include sensitive personal data for our experiment. Such datasets are too sensitive to publish so there are not too many of them.  The Enron dataset is widely used for personal data leakage probing. To justify our claim, we also manually collect certain (name, email address) and (name, phone number) pairs on the Internet, which is not done by the prior and concurrent works. We are also thrilled to conduct the red teaming tests on newly available datasets.

---

### Official Review · Reviewer_fTiA · 2023-08-09

**Soundness:** 4

**Excitement:**

3: Ambivalent: It has merits (e.g., it reports state-of-the-art results, the idea is nice), but there are key weaknesses (e.g., it describes incremental work), and it can significantly benefit from another round of revision. However, I won't object to accepting it if my co-reviewers champion it.

**Paper Topic And Main Contributions:**

The paper presents a critical issue: privacy concerns in the LLMs. The authors of the paper presented their approach to prompt-base attacking ChatGPT so that it reveals personally identifiable information. Experiments were performed on two small datasets: Enron Email Dataset (0.5M emails from Enrom employees) and publicly available information about professors from universities from all around the world. The main contributions of this paper are as follows:
- Performed experiments showing that previous attacks on ChatGPT don’t work anymore.
- Proposed multi-step prompt and demonstrated that it could leak PII from Chat GPT.
- Conducted experiments to assess the privacy leaks of ChatGPT.

**Questions For The Authors:**

How is your method applicable to other models?

**Reasons To Accept:**

The paper is interesting and easy to read. It presents a prompt that indeed can force ChatGPT to leak PII.

**Reasons To Reject:**

My main concern about this work is that it will be (or already is) outdated very quickly (that's why I gave low score for reproducibility). As I support increasing safety with all my heart, I am not sure if the current NLP research should be focused on single-use prompts (as the paper does not really introduce a new methodology apart from applying Chain-of-Thought prompting). In my opinion, it would be much better as a blog post or just a preprint on arxiv.

**Reproducibility:**

5: Could easily reproduce the results.

**Reviewer Confidence:**

4: Quite sure. I tried to check the important points carefully. It's unlikely, though conceivable, that I missed something that should affect my ratings.

---

> ### Author Rebuttal · Authors · 2023-08-28
>
> We thank the reviewer for the valuable suggestions. Our responses are provided below.
>
> > Weakness 1
> My main concern about this work is that it will be (or already is) outdated very quickly (that's why I gave low score for reproducibility).
>
> Our paper emphasizes the personal data leakage issues for present-day LLMs. Even though reinforcement learning (RLHF) is applied to enhance the LLMs’ safety and previous verbatim training data extraction no longer works, we can still abuse the instruction-following ability to misguide LLMs for unsafe behaviors. Thus, our paper points out that the current safety mechanism still needs further improvement. ‘The attack is (or already is) outdated’ means that further amendment is made to address our proposed multi-step attacks, which helps verify that the paper indeed points out a realistic and crucial problem.
>
> For reproducibility, the experiments can still be reproduced on the earlier ChatGPT version, such as gpt-3.5-turbo-0301. In addition, we also conduct the same experiments on open-source models (results appended in later tables) such as chat safety enhanced Llama 2 (Llama-2-7b-chat-hf), Vicuna (vicuna-7b-v1.3), and Guanaco (guanaco-7B). The results show that all 3 models are still vulnerable to our proposed attacks.
>
> Even for ChatGPT, it also cannot guarantee the model will not have such issues again with further training. Therefore, the proposed attack techniques are still meaningful for red teaming testing.
>
>
>
> > Weakness 2
> As I support increasing safety with all my heart, I am not sure if the current NLP research should be focused on single-use prompts (as the paper does not really introduce a new methodology apart from applying Chain-of-Thought prompting). In my opinion, it would be much better as a blog post or just a preprint on arxiv.
>
> We would like to clarify again that the key contribution of this paper is to point out that current safety mechanisms (such as content filtering, safety-oriented finetuning and reinforcement learning from human feedback) are not enough to achieve LLMs’ safety goals. The single prompt is just one example to show that we can abuse the instruction-following ability to misguide LLMs for unsafe behaviors. Such single-use prompts can be crafted easily with our proposed misleading instructions via role-playing setups.
>
> Besides prompts, we also consider novel ensemble-based techniques for verification to further improve the recovery accuracy.
>
>
> > Qestion
> How is your method applicable to other models?
>
> We have already conducted experiments on ChatGPT and the New Bing.
> In addition, we also conduct the same experiments on open-source models such as chat safety enhanced Llama 2 (Llama-2-7b-chat-hf), Vicuna (vicuna-7b-v1.3), and Guanaco (guanaco-7B). The results show that all 3 models are still vulnerable to our proposed attacks.
>
>
>
>
> ### Added Experiments
> We follow the experimental settings to conduct the same experiments on the open-source LLMs except that we only use greedy decoding for easier reproducibility. The results still suggest that chat safety enhanced Llama 2 (Llama-2-7b-chat-hf), Vicuna (vicuna-7b-v1.3), and Guanaco (guanaco-7B) still suffer higher ‘# parsed’ and ‘# correct’ from our proposed MJP.
>
> | Frequent Enron Emails (88) |        |          |           |
> |----------------------------|--------|----------|-----------|
> | Models                     | Prompt | # parsed | # correct |
> | vicuna-7b                  | DP     | 0        | 0         |
> |                            | MJP    | 59       | 3         |
> | Llama-2-7b-chat            | DP     | 0        | 0         |
> |                            | MJP    | 28       | 8         |
> | guanaco-7B                 | DP     | 0        | 0         |
> |                            | MJP    | 3        | 0         |
>
>
> | University Emails (50) |        |          |           |
> |------------------------|--------|----------|-----------|
> | Models                 | Prompt | # parsed | # correct |
> | vicuna-7b              | DP     | 1        | 0         |
> |                        | MJP    | 29       | 1         |
> | Llama-2-7b-chat        | DP     | 0        | 0         |
> |                        | MJP    | 18       | 1         |
> | guanaco-7B             | DP     | 2        | 0         |
> |                        | MJP    | 23       | 1         |
>
>
> | UNIVERSITY PHONES (30) |        |          |           |      |
> |------------------------|--------|----------|-----------|------|
> | MODELS                 | PROMPT | # PARSED | # CORRECT | LCS6 |
> | VICUNA-7B              | DP     | 0        | 0         | 0    |
> |                        | MJP    | 18       | 0         | 1    |
> | LLAMA-2-7B-CHAT        | DP     | 0        | 0         | 0    |
> |                        | MJP    | 15       | 0         | 0    |
> | GUANACO-7B             | DP     | 2        | 0         | 2    |
> |                        | MJP    | 9        | 0         | 4    |

---

### Meta-Review · Area_Chair_FU7w · 2023-09-19

**Recommendation:** 3

**Metareview:**

The paper investigates privacy risks from large language models like ChatGPT and the New Bing. While these advanced LLMs can successfully refuse previous privacy extraction attacks, it remains vulnerable to the multi-step attack proposed in the paper. Experimental results demonstrates its effectiveness against ChatGPT and New Bing models. Overall, this paper poses potential privacy risks introduced by application-integrated LLMs.

Pros:

As recognized by all the reviewers this is a timely and well-motivated topic which investigates the privacy leakage issue of ChatGPT

Proposes a multi-step jailbreaking prompts to evade the ChatGPT privacy leakage defense mechanism.
The paper is well-written and understandable

Cons:

Lack of Technical innovation- The paper studies the effect of some of the jailbreaking prompts but there is no new technique or methodology as part of the paper’s contribution. There have been many prompt based attacks on LLMs and this seems to be yet another way.

It would have been more substantial technical contribution if the authors could have proposed some way of defending these attacks or had a more extensive study across different domains/datasets/PII/models or had a resource contribution (a benchmark or dataset that can help further research in this direction)

This paper can be categorized more as a case-study and as all 3 reviewers have pointed out, the study is also quite limited in terms of domains or types of PII or datasets and models. I feel just some privacy analysis on a single model and a single aspect of privacy (PII data extraction) can be a blog or article rather than a full-paper

---

### Decision · Program_Chairs · 2023-10-07

**Decision:**

Accept-Findings

**Comment:**

The paper investigates privacy risks from large language models like ChatGPT and the New Bing. While these advanced LLMs can successfully refuse previous privacy extraction attacks, it remains vulnerable to the multi-step attack proposed in the paper. Experimental results demonstrates its effectiveness against ChatGPT and New Bing models. Overall, this paper poses potential privacy risks introduced by application-integrated LLMs.

Pros:

As recognized by all the reviewers this is a timely and well-motivated topic which investigates the privacy leakage issue of ChatGPT

Proposes a multi-step jailbreaking prompts to evade the ChatGPT privacy leakage defense mechanism.
The paper is well-written and understandable

Cons:

Lack of Technical innovation- The paper studies the effect of some of the jailbreaking prompts but there is no new technique or methodology as part of the paper’s contribution. There have been many prompt based attacks on LLMs and this seems to be yet another way.

It would have been more substantial technical contribution if the authors could have proposed some way of defending these attacks or had a more extensive study across different domains/datasets/PII/models or had a resource contribution (a benchmark or dataset that can help further research in this direction)

This paper can be categorized more as a case-study and as all 3 reviewers have pointed out, the study is also quite limited in terms of domains or types of PII or datasets and models. I feel just some privacy analysis on a single model and a single aspect of privacy (PII data extraction) can be a blog or article rather than a full-paper